# Evaluation of the Hepatitis B Vaccination Programme in Medical Students in a Dutch University Hospital

**DOI:** 10.3390/vaccines9020069

**Published:** 2021-01-20

**Authors:** Leanne P. M. van Leeuwen, Laura Doornekamp, Simone Goeijenbier, Wesley de Jong, Herbert J. de Jager, Eric C. M. van Gorp, Marco Goeijenbier

**Affiliations:** 1Department of Viroscience, Erasmus MC, University Medical Center Rotterdam, 3015 GD Rotterdam, The Netherlands; l.doornekamp@erasmusmc.nl (L.D.); s.goeijenbier@erasmusmc.nl (S.G.); w.dejong.2@erasmusmc.nl (W.d.J.); e.vangorp@erasmusmc.nl (E.C.M.v.G.); m.goeijenbier@erasmusmc.nl (M.G.); 2Travel Clinic Erasmus MC, University Medical Center Rotterdam, 3015 CP Rotterdam, The Netherlands; 3Department of Internal Medicine, Maasstad Hospital Rotterdam, 3079 DZ Rotterdam, The Netherlands; 4Department of Human Resources, Erasmus MC, University Medical Center Rotterdam, 3015 GD Rotterdam, The Netherlands; h.dejager@erasmusmc.nl; 5Department of Internal Medicine, Erasmus MC, University Medical Center Rotterdam, 3015 GD Rotterdam, The Netherlands; 6Department of Infectious Diseases, Erasmus MC, University Medical Center Rotterdam, 3015 GD Rotterdam, The Netherlands

**Keywords:** hepatitis B, healthcare workers, vaccination, long-term protection, anti-HBs

## Abstract

Healthcare workers (HCW) are at increased risk of contracting hepatitis B virus (HBV) and are, therefore, vaccinated pre-exposure. In this study, the HBV vaccination programme for medical students in a university hospital in the Netherlands was evaluated. In the first part, the effectiveness of the programme, which consisted of a vaccination with Engerix-B^®^ at 0, 1, and 6 months, was retrospectively evaluated over 7 years (2012–2019). In the second part of this study, we followed students (the 2019 cohort) who had previously been vaccinated against HBV vaccination (4–262 months prior to primary presentation) in order to investigate the most efficient strategy to obtain an adequate anti hepatitis B surface antigen titre. In the latter, titre determination was performed directly during primary presentation instead of giving previously vaccinated students a booster vaccination first. The vaccination programme, as evaluated in the retrospective first part of the study, was effective (surpassed the protection limit of 10 IU/L) in 98.8 percent of the students (95% CI (98.4–99.2)). In the second part of our study, we found that 80 percent (95% CI (70–87)) of the students who had previously been vaccinated against HBV were still sufficiently protected and did not require a booster vaccination. With this strategy, the previously vaccinated students needed an average of 1.4 appointments instead of the 2 appointments needed with the former strategy. This knowledge is important and can save time and resources in the process of occupational HBV vaccination of HCW.

## 1. Introduction

Healthcare workers (HCW) are at risk of contracting the hepatitis B virus (HBV) from infected patients. HBV is a highly contagious virus transmitted by body fluids. HCW usually obtain the virus during medical interventions, for example during needle stick injuries [1]. The clinical syndrome of a HBV infection varies from asymptomatic to fulminant liver failure; 30 percent of cases present as mild disease with fever and jaundice. HBV causes chronic disease in about 10 percent of the cases, disproportionately affecting newborns and children. Furthermore, chronic hepatitis leads to liver cirrhosis, liver failure, or hepatocellular carcinoma in approximately 25 percent of the cases [2]. Estimations indicate that in the European Union (EU), there are nearly 5 million chronic HBV cases of which 80 percent are likely undiagnosed [3]. In fact, the World Health Organisation (WHO) estimates that globally almost 90% of people chronically infected are unaware of their infection [4]. Most acute infections in Europe are sexually transmitted; however, 16 percent are caused by nosocomial transmission [5].

To mitigate this risk, HBV vaccination for HCW is recommended in all EU countries, although exact policies differ [3,6]. Lifelong protection against HBV is presumed when the concentration of antibodies against the HBV surface antigen (anti-HBs) is above 10 IU/L [7]. In 1991, the WHO recommended including HBV vaccination in national immunisation programmes (NIP) [8]. As of December 2020, 27 EU countries provide universal childhood vaccination against HBV [3]. In 2011, the Dutch government implemented this HBV vaccination initiative using a hexavalent vaccine and is now provided thrice in the first year of a child’s life. This vaccine provides protection for diphtheria, pertussis, tetanus, poliomyelitis, Haemophilus influenzae type B (Hib), and hepatitis B [9]. Before 2011, only children who fell in special risk groups (children of HBV infected mothers and children with trisomy 21) received HBV vaccination [9].

HCW without a history of HBV vaccination are requested to follow a standard HBV vaccination schedule at time points 0, 1, and 6–12 months, with a measurement of the anti-HBs titre one month after the last vaccination [10]. In less than 10 percent of HBV vaccinated individuals, no antibody response (≤10 IU/L) is seen [11]. In those cases, a second series of 3 vaccinations is usually administered with one month intervals [10]. Male gender, increased age, higher BMI, smoking, and concomitant disease are associated with this risk of non-responding [12]. In case HCW received their vaccinations at some point in the past (mostly for travel purposes), the strategies to achieve adequate anti-HBs titres differ [6]. Clinical discussions with other Dutch vaccination centres showed that some centres, including our own vaccination centre (Erasmus MC, University Medical Centre Rotterdam), administer booster vaccinations during the first visit of the vaccination clinic, while others determine the anti-HBs titres directly. After a literature search about this subject, clinical questions arose around the necessity of administering a booster vaccination and if timing of the original immunization series should impact this decision [6,8]. From August 2019 onwards, we no longer administered a booster vaccination but immediately determined the anti-HBs titre to establish the best strategy for this subgroup.

In this study, we aim to evaluate the efficacy of the HBV vaccination programme of first year medical students of the Erasmus University Medical Centre, the largest university hospital in the Netherlands, in order to ensure the best strategy for the future. First, we determined the efficacy of the vaccination policy over the past 7 years (2012–2019) after both primary HBV vaccination series and booster vaccination. Secondly, we evaluated the newly implemented policy to first test immunological memory in previously vaccinated students in order to find the best strategy to ensure protection against HBV in these future healthcare workers.

## 2. Materials and Methods

### 2.1. Study Setting

This study was conducted at the in-house vaccination clinic of a large university hospital in the Netherlands (Erasmus MC, University Medical Centre Rotterdam) responsible for the occupational vaccinations of all medical students. These students are vaccinated during their first year, in order to ensure adequate protection before starting their clinical internships. For the retrospective part of this study, we included all students who were vaccinated between May 2012 and November 2019 and reviewed their laboratory results. Students who did not complete their vaccination series, including titre determination; students with a known chronic HBV positive carrier status; and students with a known severe allergic reaction to any (component of) hepatitis B vaccination, have been excluded. Students were informed about the main side effects of vaccination or venipuncture in advance. Documentation regarding the students’ vaccination history was not recorded on individual level in the majority of cases. According to protocol, anti-HBs levels had been determined in all students to ensure protective titres [10]. When this titre was insufficient (≤10.0 IU/L) or low (≤100 IU/L), hepatitis B surface antigen (HBsAg) and anti-HB-core (anti-HBc) levels were determined additionally to exclude active HBV infection [10].

A new vaccination policy for previously vaccinated students has been implemented in our centre since August 2019. Before this date, students received a full vaccination series if they had not received HBV vaccinations before and were administered a booster when the vaccination series was completed more than 3 months before. After implementation of the new protocol, blood was collected from students who previously received a complete HBV vaccination series prior to eventual administration of booster dose. These students form a separate cohort in this study. Baseline characteristics (age, gender, date of last HBV vaccination, and type of HBV vaccination) were collected from both groups.

Due to the study design, this study was not subjected to review according to the Dutch Medical Research Involving Human Subjects Act (WMO). The study complied with the Netherlands Code of Conduct for Scientific Practice from the Netherlands Federation of University Medical Centres.

### 2.2. Vaccinations

Students presenting at our clinic are generally vaccinated with a monovalent, recombinant HBsAg vaccination named Engerix-B^®^ (GlaxoSmithKline). A full series consists of 3 doses of 20 μg HBsAg per dose was given at 0, 1, and 6 months [13]. One dose of Engerix-B^®^ (GlaxoSmithKline, Brentford, United Kingdom) was administered as booster vaccination as well. In case of non-response, another series of Engerix-B^®^ was given with intervals of 1 month (at month 7, 8, and 9 since start of first series). An additional option in case of non-response is the administration of Fendrix^®^ (GlaxoSmithKline, Brentford, UK) which consists of 20 μg HbsAg with the adjuvant AS04C. Other options for HBV immunization are Ambirix^®^ (GlaxoSmithKline, Brentford, UK) and Twinrix Adult^®^ (GlaxoSmithKline, Brentford, UK)—both combined hepatitis A and B preparations [13]. However, these combination vaccines were not routinely used for primary or booster vaccination in our clinic.

### 2.3. Laboratory Tests

Laboratory tests were conducted with a chemiluminescent microparticle immunoassay (CMIA, Architect, Abbott, Chicago, IL, USA) before 2015 and with a chemiluminescence immunoassay (CLIA-K, Liaison^®^, DiaSorin, Saluggia, Italy) from 2015 onwards. The interpretation of the anti-HBs titres does not differ between these 2 manufacturers, as these measurements are set internationally (IU/L). Both HBsAg and anti-HBc are determined qualitatively using signal-to-cutoff ratio’s (S/CO).

### 2.4. Data Analysis

Hepatitis B serology results were retrieved from the local laboratory information management system LabTrain^®^ (v3.48.1.6, Bodegro, Breda, The Netherlands). In this cohort, vaccination data was collected from the patient registry systems BaseNet and Vaccinatieregister^®^ (version 2020.07.06.220-85). IBM SPSS Statistics 25 was used for data analysis. We used descriptive statistics to calculate the means, standard deviations (SD), percentages, and confidence intervals (CI). In the second part of this study, we used chi-square tests for categorical data and Mann–Whitney U tests for nominal data. Spearman’s rank correlation tests were used to measure correlation between titres and time since last vaccination; *p*-values smaller than 0.05 were considered significant.

## 3. Results

### 3.1. Retrospective Case Series

During the study period, a total of 2925 students were vaccinated at our clinic. After excluding missing data regarding baseline characteristics, the mean age of these students was 19.5 years (2.0 SD), and 34 percent (95% CI (32–36%)) were male. In the 2922 students where HBV serology was performed, 2888 (98.8 percent, 95% CI (98.4–99.2)) demonstrated sufficient anti-HBs titres (>10.0 IU/L) after their first vaccination series or booster vaccination (Figure 1). The mean age of these responders was 19.4 years (CI 19.4–19.5) which is comparable to the mean age of the non-responders (20.4 years (19.1–21.8), *p* = 0.205). In the non-responder group, 40 percent consisted of male compared to 33 percent in the responder group (*p* = 0.346).

Out of the 34 non-responders (Anti-HBs ≤ 10.0 IU/L), 2 students (6%) tested HbsAg and anti-HBc positive. These results were reported to the Municipal Health Services as obliged by the Dutch law, and the students were referred to the appropriate healthcare facilities. Of the remaining 32 non-responders, 8 students did not show up for their second series during this study period. Of the 24 students who received a second series (7, 8, and 9 months) of Engerix^®^ (GlaxoSmithKline, Brentford, UK), 21 (88%) showed sufficient titres. The remaining three students received a booster vaccination of Fendrix^®^ (GlaxoSmithKline, Brentford, UK) after which two students responded (anti-HBs > 10.0 IU/L). The student who did not respond was referred to the occupational specialist.

### 3.2. Necessity of Booster Vaccination

From August 2019 onwards, 352 students visited the vaccination clinic. Eighty of them (22.7%) previously completed a HBV vaccination series, and blood was collected independently of the date of their last vaccination (range 4–262 months). Sixty-four (80%, CI [70%–87%]) still had a protective level of antibodies (>10.0 IU/L). In this group, the average duration between the last vaccination and the titre determination was significantly shorter compared to the group (n = 16) with an anti-HBs titre ≤ 10.0 IU/L, with 79 (95% CI, 65–94) versus 122 (95% CI, 90–153) months respectively (*p* = 0.018, Table 1, Figure 2). Gender and mean age did not differ between these groups. The direct responders were often vaccinated with the bivalent vaccines Twinrix^®^ (GlaxoSmithKline, Brentford, UK) and Ambirix^®^ (GlaxoSmithKline, Brentford, UK), whereas Engerix^®^ (GlaxoSmithKline, Brentford, UK) was given more frequently in the other group. However, no significant difference in vaccination scheme between these 2 groups was found (*p* = 0.067). A significant negative correlation (Spearman r = −0.36, *p* = 0.001) was found between the level of anti-HBs antibodies and the time elapsed between the last vaccination and the date of blood collection for HBV serology.

Although 80 percent of this cohort (the direct responders) only needed 1 appointment for blood collection, 20 percent needed 2 additional appointments: 1 for a booster vaccination and another 1 month post-vaccination for another blood collection. In order to identify the most efficient and convenient strategy, we elaborated 5 different scenarios (Table 2) using the data from this cohort. In scenario A, our policy before August 2019, all previously vaccinated students received a booster at their first appointments. In scenario B, our policy since August 2019, titre determination was performed directly in all previously vaccinated students. Booster vaccinations were only given when the anti-HBs titre was ≤10 IU/L. In the other hypothetical scenarios, policy at first appointment depended on time passed since the last HBV vaccination. For example, in scenario C, 49 students who completed their vaccination series more than 5 years before (n = 49) received booster vaccination at their first appointments. The other 31 underwent titre determination first of which 2 were insufficient. The average number of appointments per student was 1.7. We calculated the same for the cut-off at 10 years (scenario D) and at 15 years (scenario E). According to our data, the most efficient strategy to ensure sufficient protection consists of direct determination of anti-HBs titres at first appointment (scenario B). Hereby, it can also be considered to give a booster vaccination to students who completed their original series more than 15 years before, as this is just as efficient (scenario E).

## 4. Discussion

In this study, we found that almost 99 percent of the students who presented at our centre between 2012 and 2019 had protective anti-HBs titres. Furthermore, we found long-lasting protective anti-HBs titres in 80 percent of the students who completed a primary HBV vaccination series in the past. This new policy turned out to be more efficient compared to the previous policy, which dictated the administration of a booster vaccination prior to a titre check.

Adequate HBV vaccination induces a protective level of antibodies in more than 95 percent of healthy infants, children, or adolescents and is considered to provide lifelong protection [11]. Therefore, generally in immunocompetent subjects, titre administration of a booster is considered unnecessary after HBV vaccination [11,14]. In our cohort, we found a protective level of antibodies in almost 99 percent. The higher rate found in our cohort could be explained by the fact that our 7-year cohort includes not only subjects that recently completed their primary vaccination series, but also subjects receiving a booster vaccination when they previously completed their original series. As the rate of previous vaccinated students was 22.8 percent in the cohort vaccinated after August 2019, we can assume this proportion was about as high in the years before, which may explain our high rate of seroprotection.

Although the newly implemented policy will prevent many unnecessary booster vaccinations in the upcoming years, plans for Dutch occupational vaccination clinics will change in 2029. By then, the first generation of students who are HBV vaccinated under the NIP are expected to start their medical studies. In order to be prepared for this new situation, we evaluated our data and compared this to studies from countries that had implemented HBV in their NIP at an earlier stage. In 1991, Italy was one of the first countries to add HBV to their NIP [14]. Italian studies have shown that insufficient anti-HBs levels are relatively more present in individuals vaccinated during infancy than in individuals vaccinated at an older age, even when corrected for years past since vaccination [14,15,16,17]. For example, in 2 studies evaluating anti-HBs titres in young adults, 20 years after their infant vaccinations, only 32 and 37 percent showed titres above 10 IU/L [17,18]. As we and other researchers [14,17] have found that there is a negative correlation between the level of anti-HBs antibodies and the time elapsed after the last vaccination, waning anti-HBs titres are a probable explanation of these differences.

However, a lack of antibodies in the bloodstream of vaccinated individuals does not imply absence of immunity. Previous studies have shown that in subjects whose antibodies decayed, sufficient titres are detected within several days after in vitro B-cell stimulation with HbsAg [8]. Since the mean incubation time of a natural infection is 60–90 days [2], this secondary immune response will prevent these subjects from contracting a clinical relevant infection. Furthermore, HBV vaccination does also evoke a T-cell response, which has been demonstrated in previous studies both in vivo [19] and in vitro [20,21,22]. Ideally, previously vaccinated subjects with insufficient anti-HBs titres should be tested for the presence of HBV specific B- and T-cell memory without the use of a booster vaccination [20]. Nevertheless, this is a costly and time consuming method compared to the administration of a booster vaccination.

As 95 percent of vaccinated individuals normally respond to primary HBV vaccination series and the rest may rely on herd immunity, anti-HBs determination is not performed routinely in immunocompetent subjects. However, as HCW have a higher individual risk, post-vaccination antibody testing is recommended to ensure adequate immunological priming [8,23]. Future HCW, vaccinated under the NIP, will lack proof of an effective primary vaccination series, and, as such, titre determination seems inevitable. Since previous studies have shown that these titres are insufficient in more than half of the cases [17,18], switching back to the old strategy (giving booster vaccination to all students at first appointment) should be reconsidered from 2029 onwards.

This research has some limitations that have to be taken into account. First, in our 7-year cohort, the vaccination history per student was lacking. Therefore, we could not specify the number of people who did receive a full vaccination series versus the individuals who only received a booster. Second, our separate cohort of previously vaccinated students was too small to allow scientific rigor. Furthermore, their primary vaccination series were given relatively recently. Because titre determination is not routinely performed after HBV vaccination outside healthcare settings, no anti-HBs levels determined after the primary series were available of this cohort. A comparison between the peak antibody levels one month after the original series and the actual anti-HBs level could have provided more information regarding the potential of their immune response. Lastly, as policies between countries are very different [5], the results of this study are not universally applicable. However, several European countries have not yet implemented universal childhood HBV vaccination (Denmark, Finland, Iceland, and Sweden) or implemented HBV vaccination in 2017 (Norway and the United Kingdom), so our results can be of guidance for them as well [24].

In conclusion, we showed that 98.8 percent of our 7-year cohort of future HCW had protective anti-HBs titres after either a primary vaccination series for HBV or a booster vaccination. Thereby, we found that 80 percent of previously vaccinated students had sufficient anti-HBs levels without receiving a booster. With this data, we conclude that, until 2029, the most efficient strategy to ensure protection in previously vaccinated students is to directly draw blood to determine the anti-HBs level. After 2029, when individuals who were HBV vaccinated under the NIP will start studying medicine, this policy will have to be revised. Future alternative immunological methods to verify successful HBV vaccination other than anti-HBs titres could be helpful to prevent many potentially unnecessary booster vaccinations in future HCW.

## Figures and Tables

**Figure 1 vaccines-09-00069-f001:**
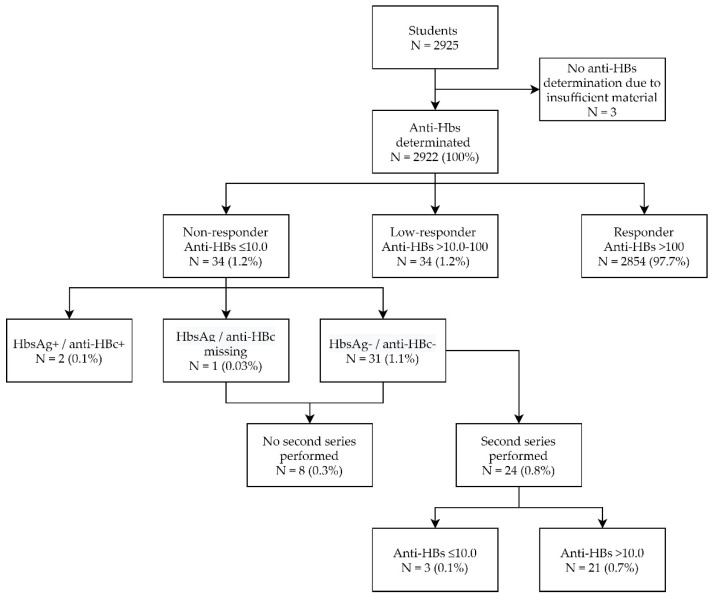
Evaluation of the occupational vaccination of students from May 2012 to November 2019.

**Figure 2 vaccines-09-00069-f002:**
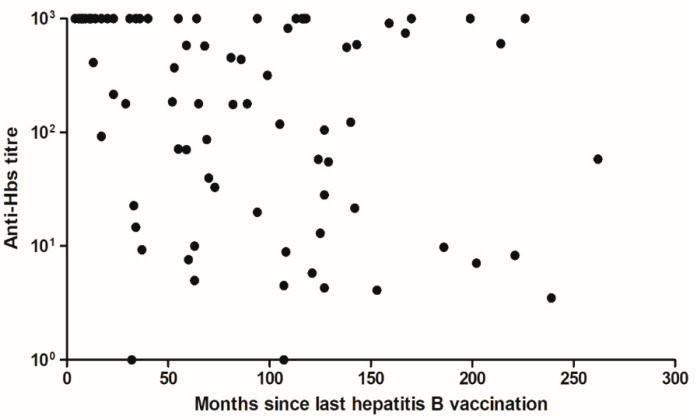
Anti-HBs titre (IU/L) as a function of months passed since the last hepatitis B vaccination. Minimum value of anti-HBs titre is ≤1.0 whereas the maximum value is ≥1000. An anti-HBs titre > 10.0 IU/L is considered protective.

**Table 1 vaccines-09-00069-t001:** Baseline characteristics of students who did not receive a booster although their last HBV vaccination was more than 3 months before.

	Titre ≤ 10 IU/L (n = 16)	Titre > 10 IU/L (n = 64)	Sign.
Age, years (SD)	18.5 (1.2)	18.4 (0.8)	0.960 ^b^
Female (%)	12 (75)	47 (73)	0.899 ^a^
Months since last HBV vaccination (SD)	122 (64)	79 (60)	0.018 ^b^
Vaccination scheme			
Ambirix^®^ (%)	8 (50)	37 (58)	0.067 ^a^
Twinrix^®^ (%)	1 (6)	8 (13)	
Engerix^®^ (%)	5 (31)	7 (11)	
Unknown (%)	2 (13)	12 (19)	

^a^ Chi-square, ^b^ Mann–Whitney U test. SD: standard deviation.

**Table 2 vaccines-09-00069-t002:** Number of booster vaccinations and/or titre determinations per appointment in 5 different scenarios.

Scenario	Appointment	1	2	3	Total	Mean nr. of Appointments Per Student
A	Booster vaccination	80	0	-	160	2
	Titre determination	0	80	-		
B	Booster vaccination	0	16	-	112	1.4
	Titre determination	80	0	16		
C	Booster vaccination	49	2	-	133	1.7
	Titre determination	31	49	2		
D	Booster vaccination	23	8	-	119	1.5
	Titre determination	57	23	8		
E	Booster vaccination	8	12	-	112	1.4
	Titre determination	72	8	12		

A. Every student receives a booster vaccination before blood is collected for a titre determination. B. Blood collection for a titre determination is directly performed in all students during first appointment. A booster vaccination is only provided to students with titres < 10 IU/L. C. Students receive a booster at first appointment if their last vaccination is more than 5 years before. Blood collection was performed directly at first appointment in the students who received their last vaccination in the previous 5 years. D. Same as C, but with a cut-off of 10 years for the last vaccination. E. Same as C, but with a cut-off of 15 years for the last vaccination.

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
