# Peer review of "Evaluation of the Hepatitis B Vaccination Programme in Medical Students in a Dutch University Hospital"

_vaccines, 2021, doi:10.3390/vaccines9020069_

Round 1
Reviewer 1 Report
The manuscript entitled “A retro- and prospective analysis of the hepatitis B vaccination programme in first year medical students in a university hospital in the Netherlands” by Leeuwen et al. reports the prospective and retrospective evaluation of hepatitis B vaccination policy. The study is providing insights for the future vaccination programme which could save time and resources. Although the study is performed in a small cohort and findings can’t be generalised but as mentioned in the MS this study could guide the future vaccination programmes.
Introduction, materials and methods and discussion sections are well written but lacking the necessary details of the effects of the vaccination in the population under study.
I have only minor comments and MS can be considered for publication after addressing these concerns.
Since the vaccination history of students were not available authors should provide data or at least comment in the discussion section about the tests performed to observe the side effects of the vaccination in the given population.
Was the liver and kidney functionality test done after vaccination? Please comment.
For how long the students that didn't required booster dose were kept under observation? Please comment.
Please discuss and refer the study performed by Brown et al. https://doi.org/10.3109/0886022X.2011.559300
Line 87: Replace settin with setting
Line 248: Spelling of Vaccines
Author Response
We thank reviewer 1 for the useful feedback. We will address the suggestions given one-by-one hereafter.
The manuscript entitled “A retro- and prospective analysis of the hepatitis B vaccination programme in first year medical students in a university hospital in the Netherlands” by Leeuwen et al. reports the prospective and retrospective evaluation of hepatitis B vaccination policy. The study is providing insights for the future vaccination programme which could save time and resources. Although the study is performed in a small cohort and findings can’t be generalised but as mentioned in the MS this study could guide the future vaccination programmes.
Introduction, materials and methods and discussion sections are well written but lacking the necessary details of the effects of the vaccination in the population under study.
Response: The effect of the vaccination in the population under study (the medical students) was measured by the determination of anti-HBs in blood serum. The results of these measurements are discussed in the result (line 149-165) and are displayed in figure 1.
I have only minor comments and MS can be considered for publication after addressing these concerns.
Since the vaccination history of students were not available authors should provide data or at least comment in the discussion section about the tests performed to observe the side effects of the vaccination in the given population.
Response: As our used vaccines are approved by pharmaceutical authorities and are extensively used for many years, specific test to observe side effects in our population were not performed. Students with a known severe allergic reaction after a previous dose of any hepatitis B-containing vaccine, or to any component of engerix-B were excluded and send to an occupation physician. To clarify this, we added this exclusion criteria to the materials and methods section.
Syncope (fainting) can occur in association with administration of vaccines. Since this is a well-known and easily treated side-effect, we have decided not to describe it specifically in our article. Other side effects of vaccination or blood drawing (muscle pain, hematoma) were communicated to the students in advance. We added this to the materials and methods section.
Was the liver and kidney functionality test done after vaccination? Please comment.
Response: liver and kidney functionality tests have not been performed after vaccination as our study population consisted of healthy young adults. In the Netherlands, it is not common practice to perform these tests in healthy individuals.
For how long the students that didn't required booster dose were kept under observation? Please comment.
Response: Follow-up after vaccination depends on the measured blood serum level of anti-Hbs antibodies. The Dutch protocol (regarding HBV vaccination in healthcare workers) states that people with a low titre (10-100) are protected for at least five years. After these five years, they might need a booster vaccination and the level of anti-HBs antibodies is determined again thereafter. Unfortunately, the current protocol does not take into account the time elapsed since vaccination. As many of our previous vaccinated students with a titer between 10-100IU/L probably had a high response (>100) more shortly after vaccination, they are now mistakenly considered as low-responder. As there is European consensus (Lancet 2000; 355: 561–65, reference nr. 11 of the manuscript) that a titre above 10IU/L is generally considered as long-term protective, these detailed information about our local protocols were deliberately omitted as this potentially confuses international readers.
Please discuss and refer the study performed by Brown et al. https://doi.org/10.3109/0886022X.2011.559300
While we think this is a very interesting article and we strongly encourage research on vaccination response in vulnerable groups, we do not think it would add value to include this article in our discussion. This article addresses a different patient group, with different types of vaccines (HBvaxpro), and discusses another issue in the field of hepatitis B vaccination.
Line 87: Replace settin with setting
We have corrected this typo. Thank you for reporting this error.
Line 248: Spelling of Vaccines
We have changed ‘vaccinees’ to ‘vaccinated individuals’.
Reviewer 2 Report
Thanks for the opportunity to review the manuscript titled “A retro- and prospective analysis of the hepatitis B vaccination 2 programme in first year medical students in a university hospi-3 tal in the Netherlands” by Leanne et al.
The authors assessed a vaccination program in fist year medical students in the Netherlands, and suggested that direct assessment of serum anti-HBs levels is the best strategy to ensure protection in previously vaccinated students.
Title
Improve language and clarity
Abstract
- I am a bit confused by the following: “In the prospective part, we investigated the most efficient strategy to obtain an adequate anti-HBs titre in students who had previously been vaccinated against HBV vaccination”. In addition to the unclear language, how is it a prospective study of individuals previously vaccinated?
- Where are the results for the “retrospective part”?
- The study setting is missing. Again, the study design is unclear.
- The section needs polishing.
Introduction
- Provide references for all important claims and avoid stacking references. There are many such statements.
- “It is estimated that there are nearly 5 million chronic HBV cases in the European Union (EU) of which an estimated 80 percent is undiagnosed”. 80% seems a lot. How does this compare with other regions of the world? Please include this information.
- Separate numbers and units (10 EI/L)
- The study aim is not clearly defined.
- Table 1 can be deleted, and the information reported in the main text.
- Please polish the section
Materials and Methods
- In Study setting (g was missing), this is called a retrospective study
- “all students of medicine and medicine-related studies” is redundant. Medical students should be fine.
- Add manufacturer information to various kits. You may want to add a paragraph describing these lab tests.
- Were there inclusion criteria? Exclusion criteria?
- Did you calculate the required sample size?
- “However, these combination vaccines were not routinely used for primary or booster vaccination in our clinic.”. What was used for boost?
- The statistical description should be improved. For example, the tests used for different variables.
- Ethics may be added to “Study design” (termed here study setting)
- Language improvement needed.
Results
- Patients characteristics should be summarized, including gender and age, also for the total study population.
- “In 27 cases, this upper limit was reached and exact determi-nation, with the use of dilution methods, was omitted. Therefore, precise values of these antibody levels are unknown.”: This is a bit strange to lose about a third of patients for such a reason.
- “Although 80 percent of this cohort (the direct responders) only needed 1 appoint-ment, 20 percent needed 2 additional appointments”: What is important in this study: number of vaccinations or appointments?
- Table 3 is a bit confusing and needs clarity
- Language improvement needed
Discussion
- In the first paragraph, summarize the findings without details.
- In the Discussion section (limitations), “retrospective design” is found again, although the title claims a prospective and retrospective design.
- Improve the English language
Author Response
We thank reviewer 2 for the useful and extensive feedback. We have tried to implement all suggestions to the best of our ability. We will address the suggestions given one-by-one hereafter.
Title
We have changed the title in order to improve language and clarity.
Abstract
- I am a bit confused by the following: “In the prospective part, we investigated the most efficient strategy to obtain an adequate anti-HBs titre in students who had previously been vaccinated against HBV vaccination”. In addition to the unclear language, how is it a prospective study of individuals previously vaccinated?
We adjusted the abstract in order to make the language more clear. Furthermore, we agree with the author that prospective is not the most suitable term for this part of the study. As with our new protocol, described in the materials and methods section, we decided to perform titre determination first instead of booster vaccination in students who previously have been vaccinated (before starting their medicine study and entering our clinic). We waited for the all the result to come back from the laboratory and invited students with an inadequate titre for booster vaccination. This is not really prospective, because we tested the titre at one point. For clearity, we have omitted ‘prospective’ from our manuscript.
- Where are the results for the “retrospective part”?
After we briefly discuss the methods of the retrospective (lines 21-23) and the second – more observational - part (23-28) of our study, we present the main results of both parts in the same order (retrospective result in lines 28-30).
- The study setting is missing. Again, the study design is unclear.
We have made arrangements to the abstract in order to make the study setting more clear.
- The section needs polishing.
We have looked critically at the manuscript and improved the clarity and the written language together with a native speaker.
Introduction
- Provide references for all important claims and avoid stacking references. There are many such statements.
We added references in the introduction section to support important claims.
- “It is estimated that there are nearly 5 million chronic HBV cases in the European Union (EU) of which an estimated 80 percent is undiagnosed”. 80% seems a lot. How does this compare with other regions of the world? Please include this information.
Indeed, it seems a lot that 80% of HBV infected people in Europe are undiagnosed. The WHO estimated a few years ago that worldwide, only slightly more than 10% are aware of their HBV infection. (https://www.who.int/news-room/fact-sheets/detail/hepatitis-b). We now include this information as well. The slightly better European numbers could well be explained by the above-average accessibility to care.
- Separate numbers and units (10 EI/L)
All numbers and units have been separated.
- The study aim is not clearly defined.
We defined our study aims more clearly in the final paragraph of the introduction.
- Table 1 can be deleted, and the information reported in the main text.
We now have deleted the table and reported the most important information in the main text.
- Please polish the section
We have looked critically at the manuscript and improved it further with the help of a native speaker to increase the readability.
Materials and Methods
- In Study setting (g was missing), this is called a retrospective study
This typo has been corrected. To minimalize the confusion between the retrospective vs prospective study design, we have made a few arrangement in this section.
2. “all students of medicine and medicine-related studies” is redundant. Medical students should be fine.
We have adjusted this to ‘medical students’.
3. Add manufacturer information to various kits. You may want to add a paragraph describing these lab tests.
We have made a new paragraph in which we describe the most important aspects of these test.
4. Were there inclusion criteria? Exclusion criteria?
We have incorporated the in- and exclusion criteria in the first paragraph of the methods section (line 94-98).
5. Did you calculate the required sample size?
An sample size calculation has not been performed. In the retrospective part of this study, our first goal was to evaluate vaccination success. As the clinic exists since 2012, data of the student cohorts since 2012 were available, so we used the maximal numbers of vaccination records as possible. In the prospective part of this study, we were dependent on the number of students that received complete HBV vaccination before first presentation in our clinic (usually due to travel reasons). A bigger sample size would have given this part definitely more power, as stated as a limitation in line 270-271.
6. “However, these combination vaccines were not routinely used for primary or booster vaccination in our clinic.”. What was used for boost?
We understand the unclearness. We added “One dose of Engerix-B® (GlaxoSmithKline) was given as booster vaccination as well.” In line 121-122
7. The statistical description should be improved. For example, the tests used for different variables.
We have made adjustments to describe the statistic section in more detail.
8. Ethics may be added to “Study design” (termed here study setting)
We have added this paragraph to ‘study setting’.
9. Language improvement needed.
Our manuscript has been additionally revised by a native English speaker.
Results
1. Patients characteristics should be summarized, including gender and age, also for the total study population.
Gender and age for the total study population were added to the results.
2.“In 27 cases, this upper limit was reached and exact determi-nation, with the use of dilution methods, was omitted. Therefore, precise values of these antibody levels are unknown.”: This is a bit strange to lose about a third of patients for such a reason.
We agree with the reviewer that a lot of data was lost by this strategy, therefore we decided to omit this analysis.
2. “Although 80 percent of this cohort (the direct responders) only needed 1 appointment, 20 percent needed 2 additional appointments”: What is important in this study: number of vaccinations or appointments?
As every appointment contains a needle stick (vaccination or titre determination), the number of appointments equals the number of needle sticks. Less appointments means less vaccinations and less titre determinations. Furthermore it consumes less time of the staff of the vaccination clinic and the students. We have made some arrangements to the manuscript to make this more clear.
3. Table 3 is a bit confusing and needs clarity
We clarified table 3. We hope this is, together with the table caption clear to the readers now.
4. Language improvement needed
Our manuscript has been additionally revised by a colleague with native English skills
Discussion
1. In the first paragraph, summarize the findings without details.
We revised the first paragraph of the discussion.
2. In the Discussion section (limitations), “retrospective design” is found again, although the title claims a prospective and retrospective design.
We understand the confusion. We have adapted the limitation section.
3. Improve the English language
Our manuscript has been additionally revised by native English speaker.
Reviewer 3 Report
A manuscript, "A retro- and prospective analysis of the hepatitis B vaccination programme in first year medical students in a university hospital in the netherlands", written by Leanne PM van Leeuwen et al., persuits two aims; 1) evaluation of HBV vaccination according to NIP in Netherlands and 2) assessment strategic options for health care workers, specifically medical students. The subject of this study is interesting and outcome of the study has value in determining most cost and time effective HBV vaccination process in HCW. Overall the manuscript was written in good flow that it was legible
Please confirm if IE/L, unit for anti-HBs antibody, is commonly used, also if so, provide full unit before using abbreviation. "mIU/ml" seems common in other papers.
Author Response
A manuscript, "A retro- and prospective analysis of the hepatitis B vaccination programme in first year medical students in a university hospital in the netherlands", written by Leanne PM van Leeuwen et al., persuits two aims; 1) evaluation of HBV vaccination according to NIP in Netherlands and 2) assessment strategic options for health care workers, specifically medical students. The subject of this study is interesting and outcome of the study has value in determining most cost and time effective HBV vaccination process in HCW. Overall the manuscript was written in good flow that it was legible
Please confirm if IE/L, unit for anti-HBs antibody, is commonly used, also if so, provide full unit before using abbreviation. "mIU/ml" seems common in other papers
We thank reviewer 3 for the useful feedback. We have checked the literature again regarding the use of IE/L, IU/L and mIU/ml. We have seen that all these units are used interchangeably. We think IU/L is the most fitting unit as this is the abbreviation for international unit so we adjusted this in our manuscript. IU/L and mIU/ml are interchangeable, but for continuity we only used IU/L.
Round 2
Reviewer 2 Report
The authors have addressed all my comments. As a result they have profoundly improved their manuscript.